# Assessment of Psychosocial Risk and Resource Factors Perceived by Military and Civilian Personnel at an Armed Forces Medical Center

**DOI:** 10.3390/ijerph22040494

**Published:** 2025-03-26

**Authors:** Alicia Bouché-Bencivinni, Vanessa Kratzien, Bruno Ballester, Mohamed Boua, Christine Jeoffrion

**Affiliations:** 1Univ. Grenoble Alpes, Univ. Savoie Mont Blanc, LIP/PC2S, 38000 Grenoble, France; aliciabencivinni@gmail.com (A.B.-B.); vanessa.kratzien@outlook.fr (V.K.); mohamed.boua@univ-grenoble-alpes.fr (M.B.); 2Service de Santé des Armées, France; brunoballester@sfr.fr

**Keywords:** psychosocial risks, occupational health, military personnel, prevention, military, civilians

## Abstract

The aim of this study is to identify the factors contributing to psychosocial risks (PSR) and the specific resources available within an Army Medical Center (AMC), with a view to developing a questionnaire adapted to this unique context. A qualitative study was carried out with 56 military and civilian employees, using semi-directive interviews. The results highlight similarities with PSR and resource factors already identified in the literature, such as workload, emotion management and salary recognition. They also reveal PSR factors specific to the activity of AMC staff, such as the need for availability and mobility, discipline and rigor, task shifting and impeded quality, as well as resource factors such as sports practice and cohesion. The results were used to develop a questionnaire, currently being validated, tailored to the context of army medical centers, with the aim of improving strategies for preventing Psychosocial Risks (PSRs).

## 1. Introduction

### 1.1. Context of the French Army, Our Research Field

The issue of Psychosocial Risks (PSRs) within the French armed forces is crucial to the health and well-being of military and civilian personnel, in compliance with Article L. 4121-1 of the French Labor Code. This legislation sets out employers’ obligations in terms of prevention, and also applies in full to the military, in accordance with law no. 91-1414 of 31 December 1991. However, measures to address the issue of PSRs in the army were introduced only after major reforms. The professionalization of the armed forces in 1996 and the internal reform of the Army Health Service (AHS) in 2011 [1] resulted in major organizational and personnel changes. These developments have had notable repercussions on staffing levels, with a significant reduction in AHS personnel since the abandonment of compulsory universal military service. Military personnel are no longer called up, but enlist on a voluntary basis, transforming the French army into an institution mainly made up of professionals under contract, in addition to civilians (Law no. 96-589, 1996).

The AHS’s independence from the military in 2011 has also altered its internal balance. From now on, medical and paramedical personnel from entities attached to the AMC are employed by the AHS, which has required a significant degree of internal reorganization, leading to increased versatility and a multiplication of ancillary tasks for these personnel [2]. This evolution has led to a refocusing of military medical practice towards direct force support, a practice closely linked to the operational contract. Despite this, the AHS must maintain sufficient versatility in order to adapt to a wide range of operational situations, while following the trend towards hyperspecialization in the public health field, thus ensuring that the highest level of professional good practice is maintained [2].

However, despite the desire to integrate PSRs into the framework of occupational risks, the lack of tools specifically adapted to the military organizational context and the paucity of studies carried out in this context make their prevention complicated. The aim of our study is to identify the factors contributing to PSRs and specific resources within an AMC, with a view to developing a questionnaire dedicated to this very specific context.

### 1.2. Psychosocial Risks in the Army

Interest in Psychosocial Risks (PSRs) has only recently gained notable attention in the media. This growing interest followed a series of suicides between 2008 and 2011 in major French companies, bringing the issue of PSRs to the forefront of societal concerns [3]. Legal rulings have further underscored the significance of PSRs, which are now recognized as inherent components of occupational risks. According to Gollac and Bodier [4], PSRs are defined as “risks to mental, physical and social health, engendered by employment conditions and organizational and relational factors likely to interact with mental functioning” (p. 49). These risks can have profound consequences on individuals’ well-being, contributing to stress, anxiety, burnout, and depression [5]. Preventing such risks is essential to preserving employees’ health and fostering performance at work. In this study, we use the term ‘resource’ to refer to elements available in the work environment that can mitigate psychosocial risks. These resources are external to the individual and include factors such as team cohesion, organizational support, and access to training. However, we acknowledge the importance of ‘protective factors’, which emphasize workers’ ability to construct their own coping mechanisms through their power to act [6]. Protective factors are more closely tied to individual agency and the capacity to adapt to challenging work conditions. This distinction highlights the interplay between organizational support (resources) and individual resilience (protective factors) in managing psychosocial risks.

In the context of the military, the issue of PSRs has become increasingly relevant due to evolving organizational and occupational demands. Bryon-Portet [7] highlights that stress levels in the execution of daily tasks have risen significantly in recent decades. Several factors contribute to this phenomenon, including professional responsibilities, demanding work schedules, and staff shortages [8]. Additionally, workload pressures can hinder the reconciliation of professional and personal life [9]. This challenge has intensified since the transition to a professional military service in 1996, which has been accompanied by various historical and legal reforms. As Girardet and Janowitz [10] observed, “the professional soldier is moving from the age of the ‘heroic leader’ to that of the ‘manager’ and technician. His traditional characteristics as a professional of violence are tending more and more to disappear” (p. 736). This transformation reflects a shift in the military identity, as soldiers’ tasks increasingly involve administrative and technical responsibilities. The redefinition of military identity has been associated with increased stress among personnel. According to Laberon et al. [11], two primary sources of stress among military personnel are the increase in administrative tasks [12] and the loss of a distinct occupational identity due to the professionalization of the armed forces [1]. The nature of military tasks—characterized by high demands, pace constraints, contradictory instructions, and long working hours—exposes personnel to various PSR factors [11]. One significant risk factor is the lack of autonomy at work, which is one of the six PSR factors identified by Gollac and Bodier [4]. In the military environment, hierarchical structures and institutional rules limit decision-making latitude, further exacerbating stress levels [13]. According to Karasek (1979, cited by Chapelle et al. [14]), decision latitude serves as a crucial resource that helps employees manage work demands and mitigate stress. Hierarchical relationships also play a key role in shaping PSRs within the military. According to Bryon-Portet [7], the hierarchical structure may influence the way errors are managed, as admitting mistakes could be perceived as undermining authority. This dynamic could potentially affect the quality of communication between superiors and subordinates, emphasizing the importance of promoting a culture of open dialogue and mutual respect to reduce stress and improve well-being. Despite these challenges, certain resources can help mitigate PSRs among military personnel. Physical activity, for instance, is a fundamental component of military life, with positive effects on health, productivity, and task performance [15]. Maintaining physical fitness prepares soldiers for operational duties, including equipment handling, load transportation, and casualty evacuation [16]. Beyond physical activity, social cohesion, solidarity, and a sense of belonging to a group represent vital protective factors [17]. These elements contribute to military personnel’s psychological well-being and resilience in the face of occupational demands [9]. Nevertheless, there is an evident lack of scientific literature focusing on the PSRs that can affect military personnel in peacetime. This is all the more the case with regard to the academic literature studying civilian personnel working in a military environment [11], such as professionals working in health services within the military.

#### PSR and Health-Related Resource Factors Identified Among Healthcare Professionals in a Civil Practice Context

Research on PSR factors among healthcare professionals has been conducted in various countries. For example, Ebatetou et al. [18] in the Congo identified work intensity and pace, value conflicts, and economic and social insecurity as key risk factors. Conversely, decision-making latitude and the quality of professional social interactions were identified as protective resources. In Quebec, Hegg-Deloye et al. [19] found decision latitude and recognition, rather than salary-related recognition, serve as resource factors. Conversely, a lack of social support from colleagues and supervisors can hinder the resolution of work-related tensions. Excessive extrinsic effort—where professionals feel they invest significant time and energy without receiving expected recognition—was also identified as a PSR factor [19]. Healthcare workers in Taiwan face numerous stressors, including work complexity, the need for specialization, and emergency situations, all of which contribute to work-related distress [20]. In Spain, Del Mar Molero Jurado et al. [21] identified three protective factors against burnout among nurses: perceived self-efficacy, social support, and emotional intelligence—defined as the ability to perceive, understand, and manage one’s own emotions and those of others. In France, Amiard et al. [22] found that low hierarchical support correlates with increased anxiety among nursing staff, while Lemoine et al. [23] identified work demands, emotional demands, lack of resources to perform quality work, limited opportunities for skill development, job insecurity, and unfavorable workplace relationships as PSR factors.

The COVID-19 pandemic has further exacerbated these challenges, as highlighted by Franklin and Gkiouleka [24] in their scoping review of psychosocial risks among healthcare workers. Their analysis of 220 studies identified four main sources of PSRs: personal protective equipment (PPE), job content, work organization, and social context. Women healthcare workers and nurses were particularly affected, reporting worse health outcomes. Similarly, Luo et al. [25] conducted a systematic review and meta-analysis on the psychological impact of COVID-19, revealing that 33% of healthcare workers experienced anxiety and 28% experienced depression. Nurses, women, and those with lower socioeconomic status were at higher risk, while access to sufficient medical resources and accurate information served as protective factors. Spoorthy et al. [26] also reviewed mental health issues faced by healthcare workers during the pandemic, identifying stress, anxiety, depressive symptoms, and insomnia as prevalent outcomes. Key risk factors included poor social support and low self-efficacy, while regular screening and multidisciplinary psychiatric support were recommended to address these challenges. In a similar vein, Barros et al. [27] explored the impact of psychosocial risks on healthcare workers’ mental health during the COVID-19 pandemic. Their study, involving 479 healthcare workers in Portuguese hospitals, identified work pace and intensity, work relationships, and emotional demands as significant predictors of anxiety and stress. Despite these challenges, healthcare workers reported experiencing joy and pleasure in their work, highlighting the importance of supportive work environments in mitigating emotional stress and promoting psychological well-being. This study underscores the need for further research to understand the less visible effects of work-health relationships and to develop targeted interventions to support healthcare workers’ mental health.

As we can see, the studies cited above, which set out to explore PSRs, are based separately on military personnel and nursing personnel. However, to the best of our knowledge, no study has combined these two populations, i.e., military personnel providing care, nor has it included civilian military personnel on this topic. The aim of our study is therefore to explore the PSR and resource factors specific to this civilian and military population working in AMCs in France, with a view to developing a context-specific questionnaire for improving prevention strategies.

## 2. Materials and Methods

### 2.1. Design

We opted for a qualitative methodology. This method seemed to us to be the most appropriate to implement, in terms of a satisfactory time cost and the possibility of better identifying risk and resource factors, on the basis of freer expression by the participants. We therefore conducted individual semi-directive interviews. To this end, we drew up an interview guide based on elements from the literature and observations of AMC staff activity. Our interview guide (Appendix A) initially included seven questions on five themes, such as the perception of professional demands, risks and the levers and brakes of activity. These themes were chosen to elicit data on work content and experience. A pre-test carried out with two military participants from different functions, a nurse and a paramedic, enabled us to check that the questions were relevant, by enabling us to obtain the expected data for each of the first five themes developed. On the other hand, we realized that a question on the link between work experience and health was missing. We therefore added an eighth question and a sixth theme: perception of the effects of work on health.

The first theme enabled us to contextualize the activity (work content, presence of responsibilities, organization of the activity) through an understanding of the function and missions (e.g., “Could you describe your function and responsibilities within the medical unit/veterinary group?”). The second theme looked at the factors which, according to staff, led to an increase or decrease in their workload (variations in activity and perceived workload) (e.g., “Are there peaks and troughs in your activity, and if so, when do they occur?”). The third theme dealt with perceived professional demands, and referred to the resources to be mobilized (physical, psychological, social or emotional) and the requisite skills, according to them, concerning the function they occupy (e.g., “What demands can you identify in your work?”). The fourth theme, related to the perceived levers and obstacles of the activity, asked about the factors contributing to the promotion or prevention of risks (e.g., “What do you appreciate about your work? What could be improved?”). This theme comprised three questions, thus enabling us to identify the risk and resource factors at the heart of our research. The fifth theme identified the risks perceived by staff in their work (e.g., What risks associated with your activity might you be exposed to?). Finally, the sixth and final theme, dealing with the effects of work on health, aimed to identify what factors either help or hinder health protection at work (e.g., “In your opinion, what impact could your work have on your health?”). The last two themes functioned together insofar as they enabled us to distinguish the probability represented by the risks posed as a result of the consequences of the latter on the health of employees.

### 2.2. Participants and Recruitment

The interviews depended on our being able to visit military bases of the AMC, and on the availability of local personnel. Our intention was to avoid interrupting their daily activities during our visit. Participants were informed about the research process during an annual meeting attended by most Medical Unit Heads (MRA) and Nurses in Charge of Medical Units (IRA). This meeting, organized with the agreement of the head of the organization, enabled the authors to present the study’s objective, methodology, and participation conditions, ensuring transparent and comprehensive information.

We spoke to 56 participants (i.e., 18.66% of the AMC’s 300 employees): 33 women (59%) and 23 men (41%). Of these participants, 47 were military personnel (out of a total of 261 military personnel at this AMC, or 18%), including 42 active military personnel serving a period of active duty, 4 military reservists and 1 military trainee, while 9 were civilians (out of a total of 39, or 23%) (see Table 1 for full sample composition). We had the opportunity to talk to 13 paramedics (out of a total workforce of 110, or 12%), 18 nurses (out of a total workforce of 72, or 25%), including 6 nurses in charge of medical units and 1 trainee nurse, 11 doctors (out of 54 in total, i.e., 20%), including 3 doctors in charge of medical units, 11 administrative staff, 8 of whom have civilian status (out of a total workforce of 33, i.e., 24%) and 3 of whom have military status (out of a total workforce of 15, i.e., 20%), as well as 2 veterinarians (out of 3, i.e., 67%) and 1 psychologist (out of 2, i.e., 50%). Of the nine civilian-status personnel, eight are administrative staff, while the ninth is a commissioned doctor, i.e., under military contract, but not subject to military requirements (since he has civilian status).

Participation in the interviews was entirely voluntary, without any hierarchical or peer pressure. Personnel were approached spontaneously during our visit, after being verbally informed of the possibility of participating in the study. Before each interview, oral consent was obtained, and participants had the option to decline without any consequences. We adopted a respectful approach, allowing participants to make themselves available according to their professional activity.

Finally, following data collection, two participants were withdrawn from our sample. The first was an active military doctor working in a medical unit, who had to leave the interview to attend an important professional meeting. We were unable to go beyond a third of our interview guide. The second was a military nursing student who was starting his internship in a medical unit on the same day. As a result, he had no knowledge of the activity he was carrying out. As a result, the interview was very brief, with no contribution to our topic of interest. In all, our sample thus comprised 54 participants, including 45 military personnel, 41 of whom were active military and 4 reservists, and 9 civilians. To preserve confidentiality and minimize social desirability bias, the interviews were not recorded. This approach aimed to foster a climate of trust by assuring participants that their statements would be collected anonymously and confidentially. The interviews were conducted face-to-face, primarily in an office provided by the unit heads, ensuring the discretion and confidentiality of the discussions.

### 2.3. Data Extraction

The interview phase lasted 6 weeks. In order to collect data, we visited 12 of the 18 entities making up the AMC, assisted by the occupational risk prevention officer (ORPO). Interviews lasted between 30 and 90 min. We did not record the interviews for two reasons. Firstly, we wanted to obtain a high level of participation. As this was based on voluntary participation and the willingness of staff to take part, offering an anonymous and confidential interview framework was reassuring, and recording the interviews in this context would have weakened the number of volunteers. On the other hand, it enabled us to avoid participants self-censoring what they had to say. It is notably more difficult to establish a relationship of trust with participants when individuals have the feeling that what they say will later be listened to and analyzed. We systematically asked for oral consent from participants before starting.

The interviews were conducted in pairs by the first two co-authors of this article, in order to compare and complement each other’s notes taken during the interview. The interviews were conducted face-to-face, mainly in an office made available to us by the branch managers. In this way, volunteer staff took it in turns to come and talk to the researchers. If this was not the case, the two researchers went to the staff’s place of work (offices, treatment room, pharmacy, etc.) and asked them if they were willing and able to spare a little of their time.

Participation depended on staff activity. In order to avoid interrupting or disrupting their work, staff were given one or two days’ notice of the researchers’ visit. As a result, staff were interviewed spontaneously, without having prepared or modified their activities in order to receive us. In that respect, we had access to the actual activity of the structure we were visiting.

### 2.4. Data Analysis

Concerning the data analysis procedure, we carried out a thematic content analysis of the corpus obtained. This enabled us to classify the comments collected into themes and sub-themes, based on selected “units of meaning”. These units could be words, sentences or paragraphs. Themes and sub-themes were not identified in advance, but were determined from the analyzed data, with the aim of responding to the problem. To achieve this, we used the free Taguette (version 1.4.0) qualitative content analysis software [28]. The first two authors conducted the interviews, transcribed the data, and performed the thematic analysis collaboratively. This approach ensured consistency and minimized potential biases in data interpretation.

The process used was as follows: after transcribing the interviews by pooling the notes taken—and as soon as possible after they had taken place, to avoid memory bias as much as possible—we inserted the 54 transcribed interviews into this software in order to highlight the units of meaning referring to homogeneous themes.

Following the inductive identification of sub-themes and themes, we organized them into a structured thematic framework (presented in tabular format). Within this framework, we systematically categorized the collected statements to ensure alignment with the emergent themes. This plan then formed the basis of our thematic analysis table, then the basis of our tool, the questionnaire. This work in pairs, and thus through collaborative discussion, enabled us to justify the relevance of each categorization by reaching mutual agreement. Having exhausted the themes, we drew up a map of PSR and resource factors and sub-factors identified (see Figure 1).

## 3. Results

Firstly, we categorized the factors on the basis of thematic analysis (Appendix B). To achieve this, we divided the occurrences into three broad categories: risk factors, resource factors and uncategorized. The occurrences in the first category evoked a difficulty or a hindrance in and/or to the activity that could lead to a health risk for the person at work. For example, the verbatim statement “I don’t always have the solution to patients’ problems, it’s not always easy to deal with that”, from the sub-theme concerning confrontation with the suffering of others relating to the theme of emotional demands, was categorized as a risk factor. In the second category, resource factors, occurrences were classified when they referred to the perceived levers or positive points of activity, thus helping to limit risks and indirectly preserve health. For example, the statement “I play sport regularly, not only to keep fit, but also to strengthen relationships” was classified as a resource factor, since the participant felt that playing sport had a double positive effect. Finally, uncategorized occurrences are those that were not connoted. For example, the verbatim statement “rigor is necessary” belonging to the sub-theme of discipline and rigor is neither connoted as a hindrance nor as a lever. Consequently, it does not belong to either risk factors or resource factors.

Secondly, we identified 9 themes and 36 sub-themes. Of the nine themes, two were categorized exclusively as risk factors: emotional demands and lack of recognition. On the other hand, professional identity was classified solely as a resource factor. Indeed, the respective sub-themes of these three themes belong exclusively to one or other of these two categories. As for the themes relating to the demands of military status and the intensity and complexity of work, their sub-themes belong to one or other of these two categories, and some even to both. Finally, the four remaining themes—meaning at work, social relationships at work, physical work environment and autonomy at work—are categorized as both risk and resource factors. Some of their respective sub-themes fall exclusively into one or other of these categories, while others, such as communication or work climate, belong to both. A synthesis of these themes and sub-themes is presented in Figure 1, which maps the main risk factors and psychosocial resources identified in this study.

### 3.1. Factors and Sub-Factors Perceived as Psychosocial Risks

#### 3.1.1. Recognition

The recognition factor was mentioned by 29 participants (54%). It represents 5% of the entire corpus of interviews. This is perceived mainly as a risk factor. Indeed, the majority of participants expressed a lack of recognition from peers, hierarchy and patients (“I have no recognition except a pat on the shoulder”), of the existence of the function in the civilian environment (“there is no equivalence of my function [paramedic] in the civilian environment”) and in terms of the salary (“the salary level is much less attractive than in the civilian environment”).

#### 3.1.2. Emotional Demands

Emotional demands were mentioned in 4% of all interviews, by a total of 29 participants (54%). In our study, this factor refers to confrontation with the suffering of others, given that the majority of staff are in contact with patients (“it’s hard to hear these stories”). It also concerns professional distance, i.e., the possible identification or projection of staff in the patient’s situation, given that they also belong to the army (“taking the patient’s problem too much to heart” could induce “emotional overload”). Lastly, this factor relates to emotional control (“good self-control: know yourself, stay calm with angry patients”; “a leader doesn’t show negative emotions”). AMC staff express the weight that this can represent on a daily basis in dealing with patients, or in exercising hierarchical responsibility.

### 3.2. Factors and Sub-Factors Perceived as a Resource

#### Professional Identity

Professional identity accounts for 1% of all interviews. Out of a total of 54 participants, 15 (28%) mentioned it. Mostly perceived as a resource, it is structured around three sub-factors: the civilian–military dual identity (“sense of inclusion”), reflecting the coexistence of civilian and military roles; shared common values (“cohesion and a team spirit specific to the army”), emphasizing collective norms; and a dual sense of belonging—both to the organization (“the institution is reassuring because the army is a little family”) and to a rewarding professional status (“rewarding military status”).

### 3.3. Factors and Sub-Factors Perceived as Both Risks and Resources

#### 3.3.1. Work Intensity and Complexity

Work intensity and complexity is the most salient factor in the thematic analysis (32% of the total 54 interviews). It was addressed by all 54 participants (100%). Of the seven sub-factors it comprises, six are perceived as risks: predictability (“he doesn’t always have a clear vision with regard to the future” and “a lot of hazards”); task shifting (“I take care of official paperwork, but normally I don’t have to”); possibility, need and access to training (“access to training is not easy: you have to be designated for training or refresher courses, and justify your interest for doing this or that training“, which is a hindrance for them; ‘we’re not trained, so we’re not competent’); workload (“they add to our workload when our work isn’t finished yet”); interruption of tasks (“a lot of task interruptions, which can generate stress”); and clarity regarding roles (“everyone must have their respective function, and not be forced to take on tasks that are not their responsibility”). As for versatility (“the advantage of the army is that you get to do a bit of everything”, which enables you to “diversify your skills portfolio”), it is perceived more as a resource factor than a risk factor. Adaptation is perceived more as a requirement than as a risk or resource factor. It is therefore not categorized.

#### 3.3.2. Military Status Requirements

Requirements relating to military status were the second most important topic discussed by participants. In fact, it represented 22% of the total content of the 54 interviews. All participants (N = 54) mentioned it. Firstly, it concerns transfers, availability and mobility, specific to military status and perceived as a risk factor (“lack of stability with regard to mobility and availability for the military personnel concerned”; the desire “that transfers be stopped”, as they impact “personal life, it’s the most burdensome”). Secondly, this factor relates to the reconciliation of private and professional life, perceived as a risk (“missions last several months and can be frequent”; “difficult to build a family life, it’s hard to have stability”). Thirdly, the allocation of responsibilities is also perceived as a risk factor (“multi-hat dimension”; “the responsibilities [that are added on]” can lead to “a loss of meaning at work”; “The more experience you have, the higher your rank, and the further away you are from care, because you have more responsibilities”). As for the practice of sport, this is perceived by the majority as a resource factor (“I practice sport regularly, not only to keep fit, but also to strengthen relationships”). Fifthly, the geographical location of the workplace is perceived as both a risk factor and a resource factor, depending on the participants and the place where they work: some appreciate “the geographical location, which makes it a good place to live”, “being able to go to the mountains is an opportunity”, while others express the view that “the geographical location of urban military establishments is not attractive”. Finally, discipline and rigor are perceived as professional requirements (“you need rigor”) and are not categorized.

#### 3.3.3. Social Relations at Work

The factor relating to social relations at work was mentioned by 52 participants (96% of all participants). It accounts for 15% of all interviews. It can represent both a risk and a resource factor, depending on how they are experienced. It is made up of four sub-factors: communication (“it’s harder to talk to the bosses”; “you see them [hierarchical superiors] all day long, it’s easy to talk to them”), listening and support (“he’s already talked about it, but nobody listens to him. It’s not taken into account, it’s not heard”; “the bosses are attentive and accessible”), cohesion and mutual support (“here, people are protective, human”) and work climate (“the work climate is based on attention and benevolence”; “when the medical unit is understaffed, there are almost permanent tensions”).

#### 3.3.4. The Meaning of Work

The factor relating to the meaning of work was discussed by 49 of the 54 participants (91%). This factor accounted for 13% of the total content of the interviews. This referred to the feeling of usefulness, mainly perceived as a resource factor. As an example, one of the participants mentions that his “work is important, rewarding and useful for the military”. It also comprises the sub-factor impeded quality and overall quality of work. One of the participants mentions that “when activity levels are intense, I have the impression of being less efficient, because I botch the work to quickly move on to something else, because it has to be done quickly”; of job satisfaction “I don’t find myself in the job of general care nurse in a medical unit”. Finally, we find the management satisfaction sub-factor. It refers to the difficulty of “managing [management], given the lack of training in personnel management”. Impeded quality of work and management are all perceived as risk factors.

#### 3.3.5. Autonomy at Work

The autonomy factor accounts for 5% of total occurrences (or appearances in the discourse). It is addressed by 37 participants (69%) and comprises four sub-factors. The first refers to forced self-training, and more specifically to “learning on the job”, i.e., “without training, developing one’s skills”. It is perceived as a risk factor. The second concerns the free organization of tasks. It is generally perceived as a resource factor. For example, some participants mention that “autonomy preserves health”, while others say they are free to organize themselves as they see fit, “I organize myself as I wish”. The third deals with temporal autonomy, perceived as both a risk factor and a resource. Some say that “there’s no time pressure”, while for others, work “rarely stops, and they’ve already stopped going to the toilet because it was too busy”. The latter refers to the possibility of self-fulfillment. For some participants, for example, this means “a lot of sharing of knowledge and experience”.

#### 3.3.6. Physical Work Environment

The physical work environment factor accounts for 2% of total occurrences. It was mentioned by 17 participants out of 54 (31%), and refers to the perception of physical working conditions, as well as the possibility of personalizing one’s working environment. According to the staff, it represents either a risk factor or a resource factor, depending on whether the physical working environment is evaluated positively (“working in new premises, in a comfortable office”) or negatively (“the equipment is outdated”).

### 3.4. Another Theme Identified: Career Transitions

The thematic analysis highlighted the theme of retention and career transitions. It is neither a risk nor a resource factor. It enables us to link the individual’s perception of their overall health with their intention to either make or not make a professional transition (e.g., to civilian life, military or civilian retraining, or other occupations). This is an interesting element to consider, as it lends weight to the importance of risk prevention—and in our case, PSRs—by alerting us to the rate of job rotation (turnover). Indeed, if several employees are planning to leave the organization, this not only raises the question of working conditions, but also their motivation to stay, which in turn raises the question of loyalty. On the other hand, it enables the head of the organization and the staff in charge of recruitment to anticipate needs and plan resources accordingly. In this sense, it contributes to the prevention of risks and represents a form of prevention by avoiding aggravating staff, or even controlling them. In this way, it draws the attention of decision-makers to the role of prevention.

## 4. Discussion

### 4.1. Main Findings

The aim of our research was to identify specific PSR and resource factors within a CMA, with a view to developing a questionnaire dedicated to this very specific context. Most of the factors and sub-factors identified in our study are supported by the literature.

Among the psychosocial risk factors supported by the literature, emotional demands were highlighted as a major PSR factor, particularly through the sub-factors of confrontation with the suffering of others, professional distance, and emotional control. These findings align with previous research showing that emotional demands in caregiving professions can contribute to stress and burnout [23]. Recognition emerged as another salient PSR factor. The lack of recognition from peers, hierarchy, and patients, as well as the absence of equivalence of certain functions in the civilian environment, contributes to feelings of frustration and job dissatisfaction. This result corroborates the findings of Letonturier [1] and Hegg-Deloye et al. [19] on the impact of recognition on occupational health. Work intensity and complexity represents one of the most prevalent PSR factors. This theme encompasses workload, task shifting, role clarity, interruptions, and access to training. Our findings are consistent with those of Pflanz and Ogle [8], who demonstrated the link between workload and psychological distress among military personnel. Additionally, Ebatetou et al. [18] identified similar risk factors among healthcare professionals.

In addition to these factors, our study highlighted eight sub-factors specific to the medical support activity for military personnel. The practice of sport, inherent to the military profession, may correspond to both a resource factor, by promoting physical and mental well-being [15,16], and a constraint when perceived as an additional obligation. The geographical location of the workplace—whether alpine or urban—can represent either a risk or a resource, depending on living conditions and access to services. In addition, transfers, availability and mobility are specific to military personnel. They are mostly perceived as a risk factor. They can be both a resource, given that they represent the core of the profession and thus the raison d’être of military personnel, and a PSR due to the consequences they entail, particularly in terms of reconciling professional and personal life. For the participants, discipline and rigor represent a professional requirement to be maintained with the aim of achieving a ‘job well done’. This sub-dimension was not rated as a hindrance or a lever, but represents a requirement perceived as important in the activity. This is why in future studies, it would be relevant to examine the respective contribution of each of these two aspects to perceived satisfaction with the work environment, in order to identify whether discipline and/or rigor constrain or enhance well-being and thus health at work. Task shifting can be linked to the assignment of responsibilities and job satisfaction. It can represent a risk factor both in terms of workload and the meaning of work. Indeed, performing a task that does not fall within one’s function seems to be quite common in the military, whether for reasons of mutual aid, obedience to one’s superior and/or lack of manpower. As a result, this can contribute to a blurred work environment, as well as a loss of meaning at work. Compulsory self-training, perceived as a psychosocial risk factor, can induce errors in the activity and possibly generate stress. The possibility of, need for and access to training were mainly identified as PSR factors by the staff we interviewed. The lack of training for certain functions, particularly team management, is a risk factor for the working climate, for instance. Indeed, the difficulty, or even impossibility in some cases, of obtaining training can lead to a lack of skills in the performance of assigned duties, a feeling of powerlessness and job dissatisfaction. This can have repercussions on the quality of work and social relations.

Finally, dissatisfaction with management–linked to the lack of managerial training–can lead to workplace tensions, with harmful effects on both staff and the organization. This finding echoes studies highlighting the importance of leadership quality in occupational health [11]. It is important to note that these nine factors do not work alone or independently. They need to be considered in relation to each other, because they are complementary. The reason being is that they contribute to generating a risk which, in turn, can lead to health consequences. They are interdependent and need to be cross-referenced in order to better identify needs, with a view to proposing tailored prevention solutions.

### 4.2. Strengths, Limitations and Research Perspectives

The value of this study is twofold. On the one hand, it enabled us to identify perceived risk and resource factors, with the aim of adapting prevention to the reality on the ground, so that preventive measures are perceived as useful for staff. In this sense, this exploratory study helped raise awareness among AMC staff of PSRs and the importance of preventing them. It also provided an opportunity to contribute to the literature on the military institution, and more specifically, on AMC personnel within their peacetime military footprint. Indeed, the literature on this subject appears to be somewhat limited in quantity and represents an issue in need of greater attention.

This study however is not without its limitations. The first concerns the representativeness of our sample. Due to time constraints, we were unable to exceed 56 interviews. On the other hand, we found that after some forty interviews, participants’ answers were redundant, suggesting data saturation. In addition, the qualitative method used may give rise to a social desirability bias on the part of participants. Nonetheless, we established a climate of trust and goodwill, specifying the anonymous and confidential nature of each interview. We explained the aim of our approach and the importance of their participation in understanding their work experience. The fact that this approach focused on the activity in a useful preventive perspective helped to limit censorship. Added to this was the fact that, as outsiders to the military institution, we had no conflicts of interest. This contributed to the participants’ freedom of expression.

A limitation of this study is the lack of feedback sessions with participants to validate the results. Due to time constraints and the terms of our contract with the institution, organizing such sessions was not feasible in the current phase. However, we have provided the institution with the complete dataset and recommendations for organizing feedback sessions in the future. This step will allow participants to engage with the findings and contribute to their validation, ensuring a more participatory approach in subsequent phases.

Finally, among the factors and sub-factors identified, two remain uncategorized: discipline and rigor, and adaptation. In concrete terms, the participants see them more as professional requirements than as a risk or a resource. This limits the possibility of proposing preventive measures for these sub-factors. In a future study, it would be useful to analyze them in greater depth to better understand their impact and explore possible actions to be taken.

### 4.3. Practical Implications/Recommendations

In terms of perspectives, the identification of various risk and resource factors has enabled us to develop a questionnaire, currently being validated, aimed at assessing these same factors on a large scale within AMCs. The decision to design a new questionnaire rather than adapt an existing tool is based on several reasons. Firstly, existing tools, such as the COPSOQ (Copenhagen Psychosocial Questionnaire, [29]) or the ERI (Effort-Reward Imbalance, [30]), although validated in civilian contexts, do not take into account the specificities of the military environment, particularly operational demands, rigid hierarchy, and the culture of ruggedness and invulnerability [7,11]. Secondly, previous evaluations conducted with the RPS-DU (Psychosocial Risks Integrated into the Single Document, INRS, [31]) revealed significant limitations, including a complex double rating system, a lack of clarity in the questions for staff, and the absence of a clear statistical analysis method. Finally, a tailor-made tool allows for the integration of factors specific to the military context, such as the notion of “multi-tasking” and task shifting, while being transversal and accessible to all personnel, both military and civilian.

The aim of this tool is to quantitatively identify the prevalence of perceived risks, and to broaden and make visible the individual and organizational strategies adopted by AMC staff. This study could provide a useful basis for the prevention of PSRs among AMC personnel, both military and civilian. Following on from this research, we recommend that the occupational risk prevention officer (ORPO) discuss aspects of work considered to be risk or resource factors, in order to co-construct prevention with the stakeholders concerned. This would help to facilitate the role of the ORPO, which would no longer be alone in designing prevention. Prevention is and must remain everyone’s business.

Secondly, it would be interesting to set up working groups to reflect on the results obtained in this study. In other words, staff who wish to participate in these groups will be able to contribute to explaining and investigating PSR factors and resources, and to co-construct prevention with the ORPO. In concrete terms, the aim will be to consider ways of limiting or even eliminating PSR factors and strengthening resource factors. To achieve this, it would be interesting to examine in these working groups how the levers identified constitute a resource in the activity, which can be mobilized or even shared (e.g., individual and organizational strategies) and how the PSR factors constitute obstacles to the achievement of quality work in addition to how they can be limited.

It would also be interesting to clarify the framework of each position, in order to limit task shifting. To this end, commanders could call on staff to jointly reflect on and define the boundaries of positions within the AMC. This would make it possible to identify the need for staff dedicated solely to administrative management activities. To meet this need, staff could be asked to set specific times or allocate a specific amount of time to administrative tasks, in order to limit interference between tasks, for example, during a medical visit with a patient.

The introduction of regular work-related discussion forums and workshops would make it possible, for example, to define what makes work meaningful, share professional practices, pool strategies and exchange experiences. During these exchanges, staff could also discuss the management dimension, enabling them to take stock of what works and what is a hindrance. However, such moments of discussion require structured facilitation by professionals such as work psychologists or ergonomists. Their role would be to ensure that the discussions remain focused on work-related difficulties and their impact on workers’ health, while fostering a constructive and participatory dialogue. This approach, inspired by activity-centered ergonomics and work psychology, would help maximize the effectiveness of these forums by providing a framework conducive to collective reflection and action. These initiatives would encourage open dialogue, strengthen collaboration and improve organizational performance. By encouraging participation and collective thinking, this would contribute to a more fulfilling and productive working environment.

Moreover, we recommend fostering a climate of trust at management level, in response to the PSR factors identified. To achieve this, as a first step, it would seem important for staff with hierarchical responsibility to be able to define their precise needs in terms of training and tools for organizing the work of a team. This could take the form of an initial meeting with the commanding officers, followed by a quarterly review of evolving needs, for example. Secondly, commanding officers could look into existing resources that could be mobilized to meet these needs (e.g., training tools, mentoring, experience sharing, analysis of practice, etc.). As a result, staff with hierarchical responsibility will be better equipped, more competent, and therefore more able to act on situations and respond to their team’s difficulties. They will be in a better position to manage their hierarchical responsibilities, which will lead to a feeling of mutual trust and a healthier working climate.

In addition, some participants suggested that, in order to compensate for network failures and maintenance that limit access to Axone software (version not specified in feedback) (the software used to access patient medical records), an offline mode should be set up to guarantee permanent access to patient records. When the connection is re-established, the software will automatically update itself to incorporate the new data. As a result, interruptions due to network failures will no longer be a hindrance to medical activity. Staff will be able to consult patient records and histories at any time, considerably improving continuity of care and facilitating medical consultations.

A lack of team spirit was also mentioned by some participants. To counter this, one of them organizes regular get-togethers with his team to strengthen ties. This facilitates communication and develops a warmer, more trusting atmosphere.

Finally, with regard to the heavy workload, one line manager indicates that he takes time at the beginning of each week to organize himself and his team. This time enables them to avoid being overwhelmed by work, and to help each other out. In the same vein, another participant uses task lists to help him organize his work. This enables him to plan and prioritize his tasks, so that he can carry them out as he goes along.

## 5. Conclusions

To our knowledge, this is the only second survey to look at the psychosocial risk factors and resources factors in the military, and the only one focusing on both military and civilian personnel in an AMC.

The items proposed in our questionnaire reflect both the problems and levers experienced or observed by staff. They incorporate a diversity of work situations, so as to include the different functions and responsibilities exercised by AMC staff, whatever their status. The key contribution of this questionnaire lies in the fact that it is innovative compared with existing tools, and thus enriches the literature.

Further work will examine its psychometric properties across all French AMCs. To achieve this, the questionnaire must be administered to two different samples, i.e., two other AMCs. Once the questionnaire has been psychometrically validated, it can be distributed nationally by the Forces Medicine Division (FMD), since it will be adapted to the specific features of each AMC. We recommend that the distribution of this tool be accompanied by prior training in its use and interpretation. To this end, it would be a good idea for the ORPO of the AMC in which we worked to take part as a trainer in a pooling meeting between ORPOs.

Furthermore, this study opens up new research perspectives. In particular, it would be useful to explore the role of resource factors and individual strategies in mitigating PSRs. Another avenue of research could be to identify the obstacles and levers to staff retention. Future research could build on this work to further explore the role of the factors highlighted in our study in the retention and attractiveness of defense professionals. This could potentially make it possible to establish a link between PSRs and staff retention, and thus reinforce the interest and place of prevention.

The results of this study have both practical and theoretical implications. The results presented here will have direct value to stakeholders in the armed forces and could inform evidence-based policies or practices around mental healthcare delivery, access, and awareness, which will improve the mental health of military and civilians going forward, as well as make the army more attractive in terms of careers.

Finally, this study further informs the relationship between work and mental health, especially in the army. Furthermore, it also underscores the need for additional research to focus on the specific psychosocial risks encountered by both military and civilian personnel who work in the army. Ensuring the military receive appropriate and frequent communication and training on mental health topics and psychosocial risks is also warranted.

## Figures and Tables

**Figure 1 ijerph-22-00494-f001:**
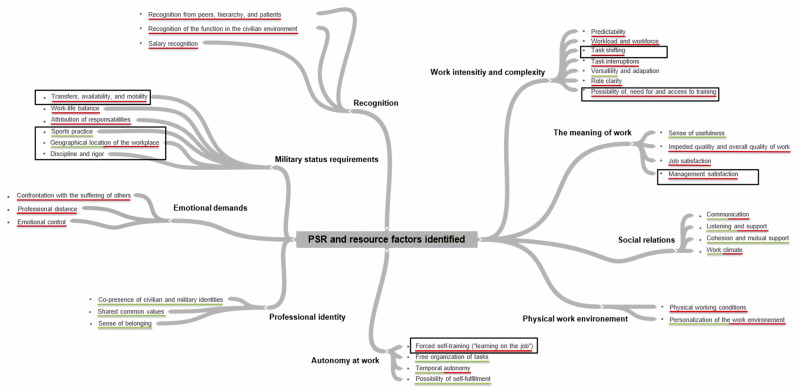
Map showing the psychosocial risk and resource factors/sub-factors identified among healthcare staff working at the AMC. Risk factors are highlighted in red, resource factors in green, and uncategorized elements are not highlighted. Factors specific to AMC staff activities are framed with rectangles.

**Table 1 ijerph-22-00494-t001:** Sample composition.

Status	N	Military Occupational Specialty	N	Rank
Civilians	9	Administrative secretaries	9	No rank
Active-duty and reserve military personnel	47	Medical auxiliaries	13	Enlisted personnel
		Administrative secretaries	2	Non-commissioned officers
		Nurses and nurses in charge of medical units	18	Non-commissioned officers
		Veterinarians	2	Officers
		Psychologist	1	Officers
		Doctors and doctors in charge of medical units	11	Officers

## Data Availability

The original contributions presented in this study are included in the article. Further inquiries can be directed to the corresponding author.

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
