# Peer review of "Assessment of Psychosocial Risk and Resource Factors Perceived by Military and Civilian Personnel at an Armed Forces Medical Center"

_ijerph, 2025, doi:10.3390/ijerph22040494_

Round 1

Reviewer 1 Report

Comments and Suggestions for Authors

Dear author,

The submitted research is relevant to the Armed Forces Medical Centre and aims to better understand psychosocial risk and resource factors. The study takes a qualitative approach to investigate the psychosocial factors affecting military and civilian employees. There are a few areas where the manuscript can be improved.

1. The objectives of the study mentioned in the manuscripts are not uniformly discussed.

2. As stated in the manuscripts, some parts of the introduction are not presented in accordance with the objectives.

3. The result part is satisfactory.

4. The discussion does not cover the psychosocial risk factors sequentially.

Wishing you the best 

Author Response

Comments 1: Dear author,

The submitted research is relevant to the Armed Forces Medical Centre and aims to better understand psychosocial risk and resource factors. The study takes a qualitative approach to investigate the psychosocial factors affecting military and civilian employees. There are a few areas where the manuscript can be improved.

Response 1: We sincerely thank you for your thorough evaluation and constructive feedback, which have significantly enhanced our manuscript.

Comments 2: The objectives of the study mentioned in the manuscripts are not uniformly discussed.

Response 2: We have addressed this comment and clarified the study objectives throughout the manuscript. The objectives are now explicitly stated in the abstract, introduction, reiterated in the methodology section, and revised in the discussion (see lines 6-8, 52-54, 185-188).

Comments 3:  As stated in the manuscripts, some parts of the introduction are not presented in accordance with the objectives.

Response 3: Thank you for your comment. The introduction has been restructured to better match the objectives of the study. We have reformulated the content of the sections based on comments made directly on the PDF manuscript you sent us as an attachment.

Comments 4:  The result part is satisfactory.

Response 4: Thank you for your positive feedback.

Comments 5:  The discussion does not cover the psychosocial risk factors sequentially.

Response 5:  Thank you very much for your valuable remark. We have tried to address your comment as thoroughly as possible by structuring the discussion to present the psychosocial risk factors in a more sequential way. Additionally, we have added references and explanations when available to better support the factors specific to the medical support activity for military personnel.

Comments 6:  Wishing you the best

Response 5:  Thank you for your kind words

Additional Note:

We also thank you for the comments provided in the annotated PDF of the manuscript. We have carefully reviewed and addressed all your suggestions, ensuring that every remark has been taken into account to improve the clarity, coherence, and quality of the manuscript.

Reviewer 2 Report

Comments and Suggestions for Authors

Methodologically and factually correct article. It contains correct and interesting research results. Correctly selected literature.

One of the few studies to look at the psycho-social risk factors and resources factors in the military, and the only one fo-cusing on both military and civilian personnel in an AMC.

Congratulaions.

Author Response

Comments 1:  Methodologically and factually correct article. It contains correct and interesting research results. Correctly selected literature.

Response 1:  We thank you for your positive feedback on the methodology, results, and literature selection.

Comments 2:  One of the few studies to look at the psycho-social risk factors and resources factors in the military, and the only one fo-cusing on both military and civilian personnel in an AMC.

Response 2:  We thank you for highlighting the unique contribution of our study in addressing psychosocial risks and resources in AMCs, particularly for mixed military-civilian staff.

Comments 3: Congratulaions.

Response 3: We thank you for your kind words and encouragement.

Reviewer 3 Report

Comments and Suggestions for Authors

Dear Author

There are a few comments:

1.     The text of the manuscript needs to be minor edited.

2.     Shouldn't the result of this qualitative study be a questionnaire with a specific cutoff point that other researchers can use?

3.     Shouldn't the explanation of Figure 1 be provided below it?

4.     Choose the keywords based on the mesh and the link below.

https://www.ncbi.nlm.nih.gov/mesh/

     Sincerely

Comments on the Quality of English Language

Dear author

The text of the manuscript needs to be minor edited.

Author Response

Comments 1:  Dear Author

There are a few comments:

The text of the manuscript needs to be minor edited. 

Response 1:  Thank you for this suggestion. Before submission, the manuscript was reviewed by a native English speaker to ensure the quality of the language. Following your comment, we have conducted a thorough re-examination of the text, rephrasing several sentences and sections to improve clarity and readability. We believe these revisions have significantly enhanced the overall quality of the manuscript.

Comments 2:  Shouldn't the result of this qualitative study be a questionnaire with a specific cutoff point that other researchers can use?

Response 2:  We thank you for this valuable remark. The primary goal of this qualitative study was to identify psychosocial risk (PSR) and resource factors specific to military and civilian personnel in Army Medical Centers (AMCs). Based on these findings, we have developed a preliminary questionnaire tailored to this context. However, this tool is not yet validated, and establishing specific cutoff points remains a necessary next step. Future work will focus on validating the questionnaire to ensure its reliability and applicability for use by other researchers and practitioners

Comments 3:  Shouldn't the explanation of Figure 1 be provided below it?

Response 3:  Thank you for your comment. We have verified and found that the explanation for Figure 1 is already placed below it. Indeed, all the text following Figure 1 details and explains the content of this figure.  However, to improve clarity, we have added a sentence that directly refers to Figure 1 just before the figure itself (see lines 366-367).

Comments 4:  Choose the keywords based on the mesh and the link below.

https://www.ncbi.nlm.nih.gov/mesh/

 Sincerely

Response 4:  We thank you for this suggestion. The keywords have been revised based on the Medical Subject Headings (MeSH) to improve the article’s visibility in scientific databases.

Comments 5:  Comments on the Quality of English Language

Dear author

The text of the manuscript needs to be minor edited.

Response 5: We have addressed all your comments to enhance the quality and clarity of the manuscript. We thank you for your constructive feedback, which has significantly improved our work.

Reviewer 4 Report

Comments and Suggestions for Authors

Dear Authors, 
Thank you for the opportunity to read your work and to contribute, through my comments and suggestions, to its possible improvement. 

My first words go to the clarity and quality of your writing, especially in the introduction section. The reader is perfectly able to follow the authors' arguments, with a solid thread and coherence in what is said.  Secondly, I would also like to highlight the bibliographical references that the authors use to support their framework on psychosocial risk factors, namely the work of Gollac and Bodier (2011) - in my view, this is an asset since psychosocial factors are often portrayed in many works as an individual issue, i.e. as a "misfit" of the worker in relation to their work context. This is not only scientifically false, but also socially harmful, because it puts the onus of responsibility on the worker. The work of Gollac and Bodier is therefore essential to counter this narrative.

Next, I detail a few points, which I encourage the authors to develop:

  • From a theoretical point of view, the authors opted to use the concept of ‘resource’ rather than ‘protective factor’. In my opinion, there is an important conceptual difference in this regard, since a ‘resource’ presupposes its availability in a given workplace.  When we talk about ‘protective factors’, this concept points more to something that is constructed by workers, depending on the room for manoeuvre granted and the demands of the job, of course, but it is a concept that directs our attention more to people's power to act (if you want to explore this topic further, I suggest reading Yves Clot's work on the notion of power to act).
  • in section 1.2.1., I would like the authors to clarify the criteria for identifying the studies mentioned. I make this comment because I can identify other studies with health professionals, demonstrating the relationship between exposure to psychosocial risk factors and musculoskeletal injuries, or mental health complaints (see 10.1016/j.shaw.2022.08.004), for example. With this comment I don't want to dispute the studies that the authors mention, but rather point out that there is a lot of literature on this subject.
  • in section 2.2., in my opinion, the authors should provide more information about the conditions of participation in the interviews. For example, could the AMC somehow know who did or didn't take part in the interviews? Relatedly, by what means were the participants able to signal that they were available to take part in the interviews? In other words, can we say that the decision to participate (or not participate) in the study was truly free and informed, without pressure from peers and/or superiors, for example?
  • in section 2.4., I think it would be important to know whether the researchers who conducted the interviews were the same ones who transcribed them and identified the themes in data analysis.
  • on page 6, Figure 1, my comment is more of a question than a suggestion. Is there any way of transforming this figure (or table) so that it's easier to identify risk factors and resources? The authors are free not to change this figure, of course.
  • in section 4.3., in my opinion, the authors should justify why they intend to design a new questionnaire, rather than adapting an existing one (adapting it, of course, to the working reality under analysis). I say this because recently there have been several questionnaires published specifically aimed at assessing psychosocial risk factors (from the well-known COPSOQ to lesser-known questionnaires such as INSAT).
  • also, in section 4.3., the authors mentioned that "it would be interesting to set up working groups to reflect on the results obtained in this study". This is a limitation of the study that should be acknowledged, since returning the data to the participants is an ethical principle in many scientific fields. Not only because of this, but also because this was an opportunity for the participants to validate the results and contribute to any possible additions. In my opinion, this reflection should be included in section 4.2.
  • in section 4.3. (the last comment, I promise), the authors suggested the introduction of regular work-related discussion forums and workshops. This is a suggestion that we often see put forward in the literature. Nothing against it. However, what is not said is that these moments of discussion about work are not “spontaneous creations”, much less are they “self-managing” moments. In other words, they are meetings (moments of debate/dialogue, if you like) that need to be mediated by a professional (a work psychologist, an ergonomist, for example) who ensures that the debate is focused on the difficulties of work and the impact these have on workers' health. If the authors consider this reflection relevant, I encourage them to read some of the works in the field of activity-centered ergonomics and work psychology in this regard (for example, Alain Garrigou, François Daniellou, to name but a few).

Once again, congratulations on your work. 

Best.

Author Response

Comments 1:  Dear Authors, 

Thank you for the opportunity to read your work and to contribute, through my comments and suggestions, to its possible improvement. 

My first words go to the clarity and quality of your writing, especially in the introduction section. The reader is perfectly able to follow the authors' arguments, with a solid thread and coherence in what is said.  Secondly, I would also like to highlight the bibliographical references that the authors use to support their framework on psychosocial risk factors, namely the work of Gollac and Bodier (2011) - in my view, this is an asset since psychosocial factors are often portrayed in many works as an individual issue, i.e. as a "misfit" of the worker in relation to their work context. This is not only scientifically false, but also socially harmful, because it puts the onus of responsibility on the worker. The work of Gollac and Bodier is therefore essential to counter this narrative.

Response 1: We sincerely thank you for your thorough evaluation and constructive comments, which have greatly contributed to the improvement of our manuscript.

Comments 2: Next, I detail a few points, which I encourage the authors to develop:

From a theoretical point of view, the authors opted to use the concept of ‘resource’ rather than ‘protective factor’. In my opinion, there is an important conceptual difference in this regard, since a ‘resource’ presupposes its availability in a given workplace.  When we talk about ‘protective factors’, this concept points more to something that is constructed by workers, depending on the room for manoeuvre granted and the demands of the job, of course, but it is a concept that directs our attention more to people's power to act (if you want to explore this topic further, I suggest reading Yves Clot's work on the notion of power to act).

Response 2: We thank you for highlighting the conceptual distinction between ‘resources’ and ‘protective factors.’ In response to your comment, we have clarified our use of the term ‘resource’ in the theoretical framework, emphasizing that it refers to external elements available in the work environment. We have also acknowledged the importance of ‘protective factors,’ which are rooted in workers’ ability to construct their own coping mechanisms through their power to act, as discussed by Yves Clot (see lines 70-79).

Comments 3: in section 1.2.1., I would like the authors to clarify the criteria for identifying the studies mentioned. I make this comment because I can identify other studies with health professionals, demonstrating the relationship between exposure to psychosocial risk factors and musculoskeletal injuries, or mental health complaints (see 10.1016/j.shaw.2022.08.004), for example. With this comment I don't want to dispute the studies that the authors mention, but rather point out that there is a lot of literature on this subject.

Response 3: We thank you for suggesting that we clarify the selection criteria for the studies cited in section 1.2.1. The studies included in this section were selected through a systematic search of relevant databases using keywords related to psychosocial risks in healthcare and military contexts. We prioritized studies focusing on emotional and organizational stressors, as these were most aligned with our research objectives. While we were not initially aware of the study you recommended (Barros et al., 2022), we have reviewed it and found it highly relevant. We have integrated this reference into section 1.2.1, along with several other studies we deemed pertinent, to broaden the scope of our analysis.

Comments 4: in section 2.2., in my opinion, the authors should provide more information about the conditions of participation in the interviews. For example, could the AMC somehow know who did or didn't take part in the interviews? Relatedly, by what means were the participants able to signal that they were available to take part in the interviews? In other words, can we say that the decision to participate (or not participate) in the study was truly free and informed, without pressure from peers and/or superiors, for example?

Response 4: We thank you for your comment on the participation conditions. We have added details to section 2.2 to clarify that participation was voluntary, confidential, and free from any external pressure. Oral consent was obtained to ensure informed participation.

Comments 5: in section 2.4., I think it would be important to know whether the researchers who conducted the interviews were the same ones who transcribed them and identified the themes in data analysis.

Response 5: We thank you for your question regarding data handling. We have clarified in section 2.4 that the same researchers conducted the interviews, transcribed the data, and performed the thematic analysis to ensure methodological consistency.

Comments 6: on page 6, Figure 1, my comment is more of a question than a suggestion. Is there any way of transforming this figure (or table) so that it's easier to identify risk factors and resources? The authors are free not to change this figure, of course.

Response 6: We thank you for your question regarding Figure 1. While we have not modified the figure itself, we have added a detailed caption to improve its clarity. The caption now explains that risk factors are highlighted in red, resource factors in green, and uncategorized elements are not highlighted. Additionally, factors specific to AMC staff activities are framed with rectangles. We believe these textual clarifications make it easier to identify and interpret the key elements of the figure (see page 8).

Comments 7: in section 4.3., in my opinion, the authors should justify why they intend to design a new questionnaire, rather than adapting an existing one (adapting it, of course, to the working reality under analysis). I say this because recently there have been several questionnaires published specifically aimed at assessing psychosocial risk factors (from the well-known COPSOQ to lesser-known questionnaires such as INSAT).

Response 7: We thank you for your question regarding the development of a new questionnaire. We have added a justification in section 4.3, explaining that existing tools do not fully address the unique challenges of military medical centers, necessitating the creation of a context-specific instrument.

Comments 8: also, in section 4.3., the authors mentioned that "it would be interesting to set up working groups to reflect on the results obtained in this study". This is a limitation of the study that should be acknowledged, since returning the data to the participants is an ethical principle in many scientific fields. Not only because of this, but also because this was an opportunity for the participants to validate the results and contribute to any possible additions. In my opinion, this reflection should be included in section 4.2.

Response 8: We thank you for highlighting the importance of returning results to participants. We have acknowledged this limitation in section 4.2, explaining that time constraints and contractual terms prevented us from organizing feedback sessions in the current phase. However, we have provided the institution with the necessary data and recommendations to facilitate this process in the future.

Comments 9: in section 4.3. (the last comment, I promise), the authors suggested the introduction of regular work-related discussion forums and workshops. This is a suggestion that we often see put forward in the literature. Nothing against it. However, what is not said is that these moments of discussion about work are not “spontaneous creations”, much less are they “self-managing” moments. Response 9: In other words, they are meetings (moments of debate/dialogue, if you like) that need to be mediated by a professional (a work psychologist, an ergonomist, for example) who ensures that the debate is focused on the difficulties of work and the impact these have on workers' health. If the authors consider this reflection relevant, I encourage them to read some of the works in the field of activity-centered ergonomics and work psychology in this regard (for example, Alain Garrigou, François Daniellou, to name but a few).

Response 9: Thank you for your suggestion regarding the facilitation of discussion forums. We made it clear in section 4.3 that these forums require professional mediation to ensure productive, health-focused discussions. We also thank you for the references you suggested.

Comments 10: Once again, congratulations on your work. 

Response 10: Thank you for your kind comments.